# Genetic Advance of Durum Wheat Under High Yielding Conditions: The Case of Chile

**Alejandro del Pozo [1],\*** , **Iván Matus [2]**, **Kurt Ruf [1,2]**, **Dalma Castillo [2]**,
**Ana María Méndez-Espinoza [1]** and **María Dolores Serret [3]**

[1]  Centro de Mejoramiento Genético y Fenómica Vegetal, Facultad de Ciencias Agrarias, Universidad de Talca, 3460000 Talca, Chile
[2]  CRI-Quilamapu, Instituto de Investigaciones Agropecuarias, 3800062 Chillán, Chile
[3]  Centre de Recerca en Agrotecnologia (AGROTECNIO), 2519825198 Lleida, Spain
\*  Correspondence: adelpozo@utalca.cl; Tel.: +56712200223

**Abstract:** In Chile, durum wheat is cultivated in high-yielding Mediterranean environments, therefore breeding programs have selected cultivars with high yield potential in addition to grain quality. The genetic progress in grain yield (GY) between 1964 and 2010 was 72.8 kg ha$^{-1}$ per year. GY showed a positive and significant correlation with days to heading, kernels per unit ground area and thousand kernel weight. The gluten and protein content tended to decrease with the year of cultivar release. The correlation between the $\delta^{13}$C of kernels and GY was negative and significant ($-0.62$, $p < 0.05$, for all cultivars; and $-0.97$, $p < 0.001$, excluding the two oldest cultivars). The yield progress (genetic plus agronomic improvements) of a set of 40–46 advanced lines evaluated between 2006 and 2015 was 569 kg ha$^{-1}$ per year. Unlike other Mediterranean agro-environments, a longer growing cycle together with taller plants seems to be related to the increase in the GY of Chilean durum wheat during recent decades.

**Keywords:** agronomic traits; carbon isotope; days to heading; grain quality; yield components

## 1. Introduction

Durum wheat (*Triticum turgidum* L. ssp. *durum*) covers ~17 million hectares worldwide, which is less than 10% of the total wheat area. However, its importance for human consumption is very high because it is used for making pasta, couscous, burghul and firik [1]. According to the International Grain Council, the largest producers of wheat in the world are the European Union, Canada, the United States, Turkey and Algeria.

For the production of high-quality durum wheat, dry environments are necessary, with warm days and cold nights during the growing season so that large grains are obtained with yellow color, vitreous kernels (more than 95%), hard texture and high test weight (about 82 kg hL$^{-1}$), alongside high protein content (greater than 10%) and strong gluten (greater than 30% wet gluten), which gives elasticity to dough for industrial use [2]. In Chile, durum wheat is grown in Mediterranean climate environments from the Valparaíso Region (32 °S) to the Biobio Region (37 °S), but mostly under irrigation conditions or in areas where rainfall is sufficient to satisfy most or all of the crop potential evapotranspiration. The sowing area has increased from 9600 ha in 2001 to 27,000 ha in 2015, and the average yield for 2011–2015 was 6.7 Mg ha$^{-1}$ [3].

Wheat yields in different regions of the world have increased greatly since the 1960s as a result of genetic improvement and better agronomic practices [4]. With the Green Revolution, breeding programs have seen the introduction of semi-dwarfing genes that interfere with the action or production of gibberellin [5], leading to a reduction in plant size and an increase in the partitioning of the above-ground

biomass towards spikes and grains [6,7]. In bread wheat, the genetic gain in grain yield (GY) was positively correlated with harvest index and the number of grains per spike and per m$^2$ [6,8]. In durum wheat, a Spanish study conducted with 12 cultivars from Italy and 12 from Spain released between 1930 and 2000 showed that the changes in grain yield were also associated with increases in the harvest index and number of grains per m$^2$ [9]. Another study carried out on 14 cultivars released in Italy between 1900 and 2000 indicated that the total aerial biomass had not changed and that the increase in yield was associated with a reduction in plant height and an increase in the harvest index and number of grains per m$^2$ [10]. These studies in durum wheat have been conducted under rain fed conditions and yields were below 6 Mg ha$^{-1}$, however, there is no information about the genetic progress in durum wheat in high yield potential environments (>10 Mg ha$^{-1}$) and how grain quality traits have been affected. Moreover, these works have not focused on studying exclusively the trends in breeding advances of post-Green-Revolution (i.e., semi-dwarf) durum wheat cultivars during the last half-century. This is despite the importance of this issue in the context of climate change and the fact that at least for bread wheat, there are studies reporting a stagnation in yields (or at least a drastic decrease in genetic advance) during the last decade [11].

In bread wheat, grain protein content, sedimentation value and wet gluten have increased in modern cultivars [8,12–14]. In durum wheat, modern dwarf and semi-dwarf cultivars have a higher gluten index compared to landraces or traditional Mediterranean cultivars [15]. Subira et al. [16] also reported significant changes in grain quality traits in a historical series of 24 durum wheat cultivars released in Italy and Spain in different periods of the 20th Century, particularly in gluten strength, sedimentation index and yellow color index. High protein content and 'strong' gluten are necessary to process semolina into a suitable final pasta product.

Physiological changes associated with breeding advances have also been reported for bread and durum wheat. For instance, modern bread wheat presented higher stomatal conductance (on an area basis) and carbon isotope discrimination ($\Delta^{13}$C; or a lower carbon isotope composition, $\delta^{13}$C), and lower oxygen isotope composition ($\delta^{18}$O) than older varieties [8,17–19]. In durum wheat, modern varieties have higher $\Delta^{13}$C (or lower $\delta^{13}$C) compared to landraces [20–22], although no clear differences were found for $\delta^{18}$O [21]. However, no information exists for durum wheat growing in a high-yielding Mediterranean environment.

The aim of this work was to analyze a) the changes in agronomic traits, grain quality and isotope composition in a set of ten durum wheat cultivars released in Chile between 1964 and 2010; and b) the progress in grain yield, plant height and test weight in selected advanced lines from the Instituto de Investigaciones Agropecuarias (INIA)-Chile breeding program. The experiments were conducted in a high-yielding Mediterranean environment between 2006 and 2015.

## 2. Materials and Methods

### 2.1. Experimental Site, Plant Material and Growing Conditions

The experiments were conducted at the Santa Rosa experimental field station (36°32′ S, 71°55′ W; 220 m.a.s.l.) of the Centro Regional de Investigación (CRI)-Quilamapu, Instituto de Investigaciones Agropecuarias (INIA). The climate corresponds to a humid Mediterranean type. During the experimental period (2006–2015), the monthly minimum average temperature was 3.1 °C (July) and the maximum 29.6 °C (January), and the average annual precipitation was 903 mm (Supplemental Table S1). The soil was a sandy loam, humic haploxerands (Andisol). Soil chemical characteristics of the top 10 cm were: pH 6.0, 8.87 mg kg$^{-1}$ of N-N0$_3$; 17.05 mg kg$^{-1}$ of P (Olsen), 0.45% of N-total, 4.5% of C and 0.33, 5.75, 0.65 and 0.48, cmol kg$^{-1}$ of available K, Ca, Mg and Na, respectively [23].

Two different experiments were conducted. In the first experiment, ten cultivars released by the INIA breeding wheat program from 1964 to 2010 (Table 1) were evaluated during three consecutive seasons (2010 to 2012). The INIA cultivars derive from germplasm introduced from the International Maize and Wheat Improvement Center (CIMMYT) and probably all of them have the *Rht-B1* gene.

**Table 1.** Cultivars of durum wheat released by the wheat breeding program of the Chilean INIA between 1964 and 2010.

| Cultivar | Year [1] | Cross/Pedigree |
|---|---|---|
| Alifén | 1964 | CAPELLI//ST 464 |
| Quilafén | 1970 | YT54/Nl08//LD 357/2 *TC |
| Chagual INIA | 1986 | 2156 3/AA" S"//PG" S" |
| Chonta INIA | 1990 | FRIGATTE"S"//RUFF/FLAMINGO"S" |
| Licán INIA | 1990 | RUFF "S"/FG"S"//MEX/3/SHWA"S" |
| Llareta INIA | 1997 | D67.54.4.9A//JORI'S'/ROSNER DURUM 119-200-4Y/3/ SAHEL77 |
| Guayacán INIA | 1997 | ALTAR84/STINT"S"//SILVER |
| Corcolén INIA | 2002 | ALGA"S"/3/CANDEALFENS5/FLAMINGO"S"//PETREL"S"/ 4/CHURRILLA"S"/5/AUK"S"/6/RUFF"S"/FLAMINGO"S"// FLAMINGO"S"/CRANE"S"/3/YAVOROS 79/HUITLES"S" |
| Lleuque INIA | 2009 | YEL"S"/BAR"S"/3/GR"S"/AFN//CR"S"/5/DOM"S"//CR"S"*2/ GS"S"/3/SCO"S"/4/HORA/6/LAP76/GULL"S"/7/LICAN |
| QUC 3104–2005 [2] | 2010 | ALTAR84/ALD"S"//STN"S"/CHEN"S"/ALTAR84/4/ATES1D |

[1] Year of cultivar release; [2] experimental line.

The experimental design was a complete block with four replications. Each plot consisted of five rows of 2.5 m length and 0.2 m apart. Sowing dates were in August of each year and the sowing rate was 220 kg ha$^{-1}$. Fertilization consisted of 1.5 t ha$^{-1}$ of lime (88%–90% $CaCO_3$) before sowing, 260 kg ha$^{-1}$ of diammonium phosphate (46% $P_2O_5$, 18% N), 200 kg ha$^{-1}$ of potassium magnesium sulfate (22% $K_2O$, 18% MgO, 22% S), 90 kg ha$^{-1}$ of potassium chloride (60% $K_2O$), 10 kg ha$^{-1}$ of boronatrocalcite and 3 kg ha$^{-1}$ of zinc sulfate (35% Zn) at sowing. After sowing, an extra 133 kg ha$^{-1}$ of urea (46% of N) was applied at tillering initiation (Zadoks 20; [24]) and 201 kg ha$^{-1}$ at the first node (Zadoks 31). Plots were furrow irrigated according to the needs of the crop (3–4 irrigations of ~50 mm each, per season). Weeds were controlled using the pre-emergence herbicide Bacara Forte 360SC, Bayer Crop Science (800 mL ha$^{-1}$; 12:12:12% w/v a.i. of flufenacet/flurtamone/diflufenican) and the post-emergence Ajax, Anasac, Chile (10 g ha$^{-1}$; 50% w/w a.i. of metsulfuron-methyl) and MCPA 750 SL, Anasac, Chile (800 mL ha$^{-1}$; 95% w/v a.i. of 2-methyl-4-chlorophenoxyacetic acid)). Since the oldest cultivars showed susceptibility to rust (*Puccinia striiformis* and *Puccinia triticina*), two applications were made of the foliar fungicide Juwel-Top, Basf (100 mL ha$^{-1}$; 12.5:12.5:15% a.i. of kresoxim-methyl/epoxiconazole/phenopropimorph). These applications were made before symptoms appeared, to avoid any interference of these diseases in the development of the plants.

In the second experiment, a selection of 46 advanced lines (F6–F8) of durum wheat from the breeding program (Durum Yield Nursery) and four check cultivars (Llareta-INIA, Corcolén-INIA, Lleuque-INIA and Queule-INIA) were tested each year from 2006 to 2015. Two trials of 25 genotypes each, including check cultivars, were established each year in an $\alpha$-lattice design with five incomplete blocks per replicate, each block containing five genotypes. There were four replicates per genotype. The plots consisted of five rows of 2 m length and 0.20 m between rows. The seed rate was the equivalent of 220 kg ha$^{-1}$. The sowing date was August of each year. Crop fertilization and weed control were as recommended for each year. Plots were furrow irrigated according to crop need (3–4 irrigations of ~50 mm each, per season). These trials were regularly conducted by the breeding program in order to test the most promising advance lines in comparison with the commercial cultivars (check cultivars); those advance lines with outstanding performance were evaluated for more than one year, and the rest were replaced by new ones. As a consequence, the set of advance lines evaluated in each year was composed of different elite genotypes.

## 2.2. Agronomic Traits

In Experiment 1 the following traits were evaluated: (a) Days from emergence to heading (DH) through periodic observations (twice per week), when approximately half of the spikes in the plot had already extruded; (b) the number of spikes per m$^2$ (SM2) by counting the spikes in a 1.0 m length

of a row; (c) the harvest index (HI), determined from a sample from the 1.0 m row at maturity and calculated as the ratio of grain dry weight to total above ground dry weight; (d) the number of kernels per spike (KS) and thousand kernel weight (TKW) from 25 spikes taken at random from each plot and (e) the number of kernels per $m^2$ (KM2) calculated as SM2 × KS. In Experiments 1 and 2, plant height (PH) from the ground to the top of the spike, excluding awns, was measured at maturity, and GY was assessed by harvesting 2 $m^2$ (five rows, 2 m long).

## 2.3. Grain Quality

The test weight was evaluated in Experiments 1 (2010) and 2 (2006 to 2015), in samples of wheat free of impurities (obtained from each genotype and replicate) using a 250 cc Schopper scale (Louis Schopper, Germany). In addition, grain samples obtained from the genotypes and replicates evaluated in Experiment 1 (in 2010) were ground in mill for wet gluten and protein content determination. Wet gluten content was determined according to the International Approved Methods of Analysis (AACCI Method 38–12.02) in 10 g of pure flour mixed with 5.5 mL of a 2% saline solution, which was homogenized and then placed in a gluten washer (Glutomatic® 2200, Perten Instruments, USA) for 5 min; then the wet gluten was weighed and expressed as a percentage of the amount of pure flour. Protein content (%) was also determined in ground grain samples placed in a quartz cuvette and the reflectance spectrum between 800 and 2500 nm was determined using near infrared reflectance spectroscopy (NIRS), Bruker, USA. Yellow berry incidence was assessed on 100 g of kernels, separating and weighing the affected grains and then expressed in percentage.

## 2.4. Total N Content and C and N Isotope Analyses

Measurements were performed in mature grains harvested in 2011 (Experiment 1). The total N content was analyzed using an elemental analyzer (Flash 1112 EA; ThermoFinnigan, Bremen, Germany). The stable carbon ($^{13}C/^{12}C$) and nitrogen ($^{15}N/^{14}N$) isotope ratios of the same mature grains were determined in the same elemental analyzer coupled with an isotope ratio mass spectrometer (Delta C IRMS, ThermoFinnigan, Bremen, Germany). Nitrogen was expressed as a concentration (g N per g of dry weight) and atropine was used as a system check in the elemental analyses of nitrogen. The $^{13}C/^{12}C$ ratios of plant material were expressed in δ notation: $δ^{13}C = (^{13}C/^{12}C)$ sample/$(^{13}C/^{12}C)$ standard – 1, where 'sample' refers to plant material and 'standard' of known $^{13}C/^{12}C$ ratios. The $^{15}N/^{14}N$ ratios were also expressed in δ notation ($δ^{15}N$) using international secondary standards of known $^{15}N/^{14}N$ ratios. More details are described in del Pozo et al. [8]. Measurements were performed at the Scientific Facilities of the University of Barcelona.

## 2.5. Data Analysis

Complete block analysis of variance (ANOVA) were performed for the set of cultivars evaluated in Experiment 1 using IBM SPSS Statistics software (SPSS Inc, USA). In addition, correlation analyses were performed between the year of cultivar release and agronomic, grain quality and isotope composition traits, and among the different traits. Trends for grain yield, plant height and test weight of 46 advanced lines and cultivars evaluated from 2005 to 2015 in Experiment 2 are also presented.

## 3. Results

### 3.1. Agronomic Traits in Cultivars Released During the Past Six Decades

Days to heading differed significantly among cultivars and also the year × cultivar interaction was significant (Table 2); it reduced in the 1990s, but then increased in the 2000s (Figure 1A). Plant height was significantly ($p < 0.001$) reduced from 108 cm in the 1960s to 90 cm in the 1970s, with a slight increase in 2010 (Table 2; Figure 1B).

**Table 2.** Mean sum of squares of the analysis of variance (ANOVA) for agronomic traits of ten durum wheat cultivars cultivated during three growing seasons (2010–2012).

| Source of Variation | d.f. | DH | GY | PH | SM2 | KS | KM2 | TKW | HI |
|---|---|---|---|---|---|---|---|---|---|
| Year | 2 | 238.9 | 187.9 | 418.1 | 413,645 | 420.6 | $345.5 \times 10^6$ | 63.5 | 0.017 |
| Cultivar | 9 | 84.2 | 20.3 | 456.7 | 36,213 | 287.0 | $50.4 \times 10^6$ | 292.5 | 0.019 |
| Block | 3 | 0.7 | **9.0** | 6.9 | 5219 | 16.3 | $19.5 \times 10^6$ | 0.2 | 0.001 |
| Year × Cultivar | 18 | *2.7* | *3.0* | 12.1 | 16,296 | *29.6* | $20.1 \times 10^6$ | *7.1* | 0.001 |
| Residual | 87 | 0.4 | 1.7 | 10.3 | 5040 | 11.5 | $12.8 \times 10^6$ | 2.9 | 0.001 |
| Total | 120 | | | | | | | | |

Level of significance is indicated in bold ($p < 0.01$) and cursive ($p < 0.05$). DH: Days to heading; GY: Grain yield; PH: Plant height; SM2: Number of spikes per m$^2$; KS: Kernels per spike; KM2: Kernel number per m$^2$; TKW: Thousand kernel weight; HI: Harvest index.

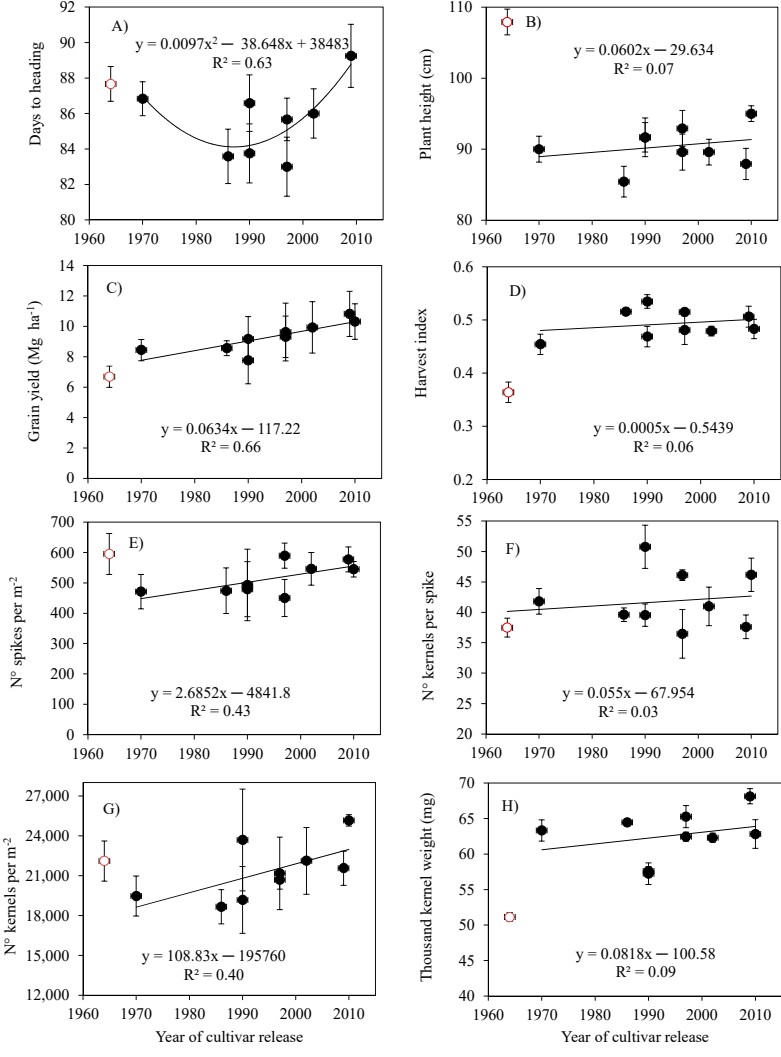

**Figure 1.** Relationships between the year of release of ten durum wheat cultivars and: Day to heading (**A**), plant height (**B**), grain yield (**C**), harvest index (**D**), number of spikes per square meter (**E**), number of kernels per spike (**F**), number of kernels per square meter (**G**) and thousand kernel weight (**H**). Values correspond to the average (±SE) of three growing seasons (2010–2012) except for HI, which was determined in 2010 and 2011. The oldest (1964) cultivar (open circle) was not considered in the regressions. Mean values of cultivars for each year of evaluation are shown in Supplemental Table S2.

GY exhibited a positive and linear relationship with the year of cultivar release ($R^2 = 0.66$; $p < 0.001$), this analysis excluding the oldest cultivar (1964; Figure 1C). The rate of increase in GY after 1960 was 72.8 kg ha$^{-1}$ per year, and excluding the oldest cultivar it was 63.4 kg ha$^{-1}$ per year. The SM2 of the ten cultivars ranged between 450 and 595 and increased significantly ($R^2 = 0.43$; $p < 0.05$) with the year of release (Figure 1E). The HI was 0.36 in the 1960s and increased to 0.45–0.53 in the 1970s and onwards, whereas TKW was 51.1 g in the 1960s and rose to 57–68 g after the 1970s, but neither trait was correlated with the year of cultivar release (Figure 1D,H). Similarly, KS was not correlated with the year of release (Figure 1F), but KM2 increased significantly ($R^2 = 0.40$; $p < 0.05$) with the year of cultivar release (Figure 1G).

The correlation matrix among the agronomic traits of the 10 cultivars evaluated during three growing seasons indicated that days to heading exhibited a positive and significant correlation with GY ($p < 0.05$) and KS ($p < 0.01$), and GY showed a positive and significant correlation with KM2 ($p < 0.05$) and TKW ($p < 0.001$; Table 3). Plant height was not correlated with GY. However, plant height showed a negative and highly significant ($p < 0.001$) correlation with TKW and HI. SM2 had a positive correlation with KM2 but a negative correlation with KS.

**Table 3.** Correlation matrix among agronomic traits evaluated in ten cultivars during three growing seasons (2010–2012).

|  | DH | GY | PH | SM2 | KS | KM2 | TKW | HI |
|---|---|---|---|---|---|---|---|---|
| DH | 1.00 | | | | | | | |
| GY | 0.44 * | 1.00 | | | | | | |
| PH | 0.36 | 0.08 | 1.00 | | | | | |
| SM2 | −0.30 | 0.08 | 0.20 | 1.00 | | | | |
| KS | 0.50 ** | 0.30 | 0.09 | −0.61 *** | 1.00 | | | |
| KM2 | 0.06 | 0.39 * | 0.33 | 0.76 *** | 0.03 | 1.00 | | |
| TKW | 0.14 | 0.59 *** | −0.55 *** | −0.03 | 0.01 | 0.01 | 1.00 | |
| HI | −0.21 | 0.02 | −0.75 *** | −0.32 | 0.34 | −0.11 | 0.51 * | 1.00 |

*: $p < 0.05$; **: $p < 0.01$; ***: $p < 0.001$ DH: Days to heading; GY: Grain yield; PH: Plant height; SM2: Spike number per m$^2$; KS: Kernels per spike; KM2: Kernel number per m$^2$; TKW: Thousand kernel weight; HI: Harvest index.

*3.2. Grain Quality and Kernel Isotope Composition in Cultivars Released During the Past Six Decades*

The test weight increased curvilinearly with the year of cultivar release (Figure 2A). The gluten and protein content tended to decrease with the year of cultivar release, although the correlations were not significant (Figure 2B,C). Yellow berry was higher in two cultivars, but there was no clear pattern with the year of cultivar release (Figure 2D).

The relationships between the year of cultivar release and N concentration or $\delta^{15}$N in kernels were not significant (Figure 3A,B). The $\delta^{13}$C of kernels tended to decrease with the year of cultivar release, although the correlation was not significant (Figure 3C). In addition, $\delta^{13}$C was negatively correlated ($r = -0.62$; $p < 0.05$) with GY, but $\delta^{15}$N was not correlated ($r = 0.03$; $p > 0.05$).

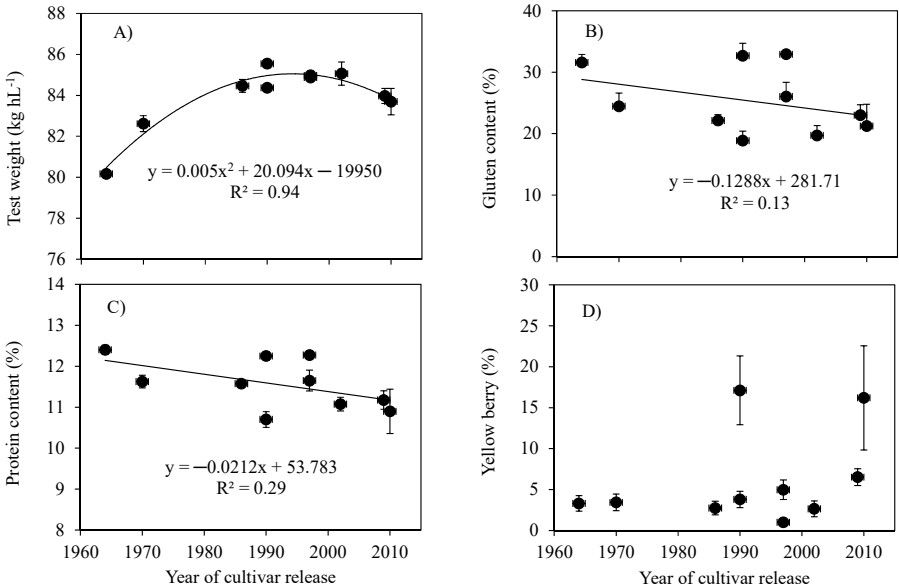

**Figure 2.** Relationships between the year of release of 10 durum wheat cultivars and kernel test weight (**A**), wet gluten content (**B**), protein content (**C**) and yellow berry (**D**), determined in 2010. Values correspond to the average (±SE) of four replicates.

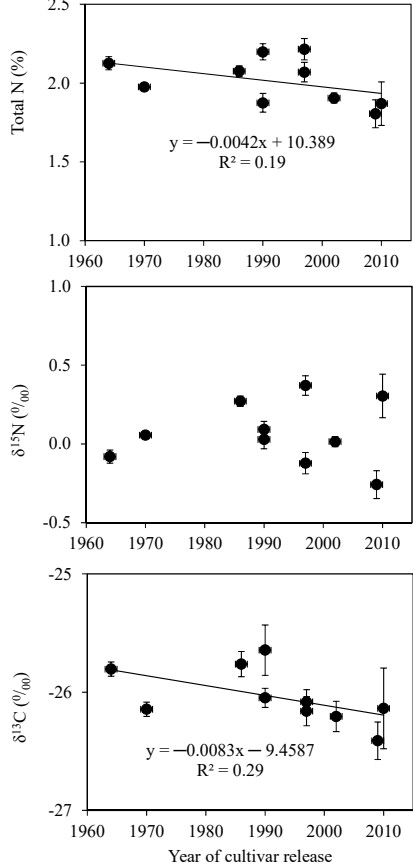

**Figure 3.** Relationships between the year of release of 10 durum wheat cultivars and the (**A**) total nitrogen, (**B**) natural abundance of $^{15}N$ ($\delta^{15}N$) and (**C**) carbon isotope composition ($\delta^{13}C$) in kernels, determined in 2011. Values correspond to the average (±SE) of four replicates.

### 3.3. Agronomic and Grain Quality Traits in Advanced Lines During the Last Decade

The GY and plant height of advanced lines increased from 2006 to 2015, reaching a maximum in 2011 with averages of $12.7 \pm 0.8$ Mg ha$^{-1}$ and $96 \pm 4.2$ cm, respectively (Figure 4A,B). GY was highly correlated with plant height ($r = 0.85$; $p < 0.001$). The check cv. 'Corcolén' followed a similar trend to the advanced lines. The average GY of advanced lines and cultivars had a positive and significant ($R^2 = 0.50$; $p < 0.001$) relationship with the year of evaluation; the regression analysis indicated that the rate of increase in GY between 2006 and 2015 was 569 kg ha$^{-1}$ per year. The test weight did not increase during this period (Figure 4C). No significant ($p < 0.05$) correlation was found between GY of advanced lines and the average temperature (maximum, minimum or mean) for the wheat growing season (August–January) from 2006 to 2015.

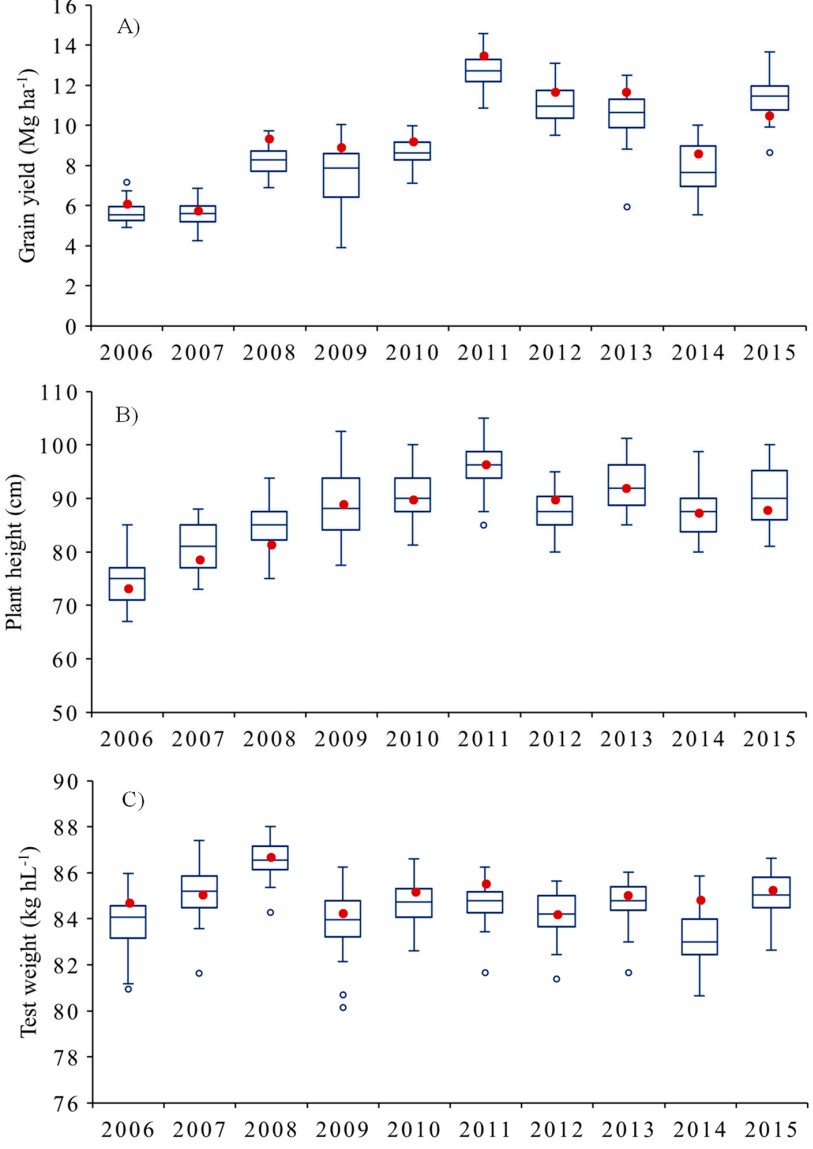

**Figure 4.** Grain yield (**A**), plant height (**B**) and test weight (**C**) for 40–46 advanced lines and cultivars of durum wheat grown under full irrigation in Santa Rosa, from 2006 to 2015. Box and whisker plots show the population minimum, 25th percentile/median/75th percentile and maximum. The open symbols indicate outlier data and the closed symbols indicate the check cultivar 'Corcolén'.

## 4. Discussion

### *4.1. Agronomic Traits*

Modern cultivars of spring durum wheat from Chile have a very high yield potential (~13 Mg ha$^{-1}$) in a Mediterranean environment, under fully irrigated conditions. The yield potential achieved in Chile is clearly higher than values recorded in the Mediterranean basin. For example, high-yielding conditions in Spain usually do not surpass 8 Mg ha$^{-1}$ [25,26], which is clearly lower than those achieved in the Mediterranean conditions of Chile. The high-yielding conditions in Spain usually imply several irrigations per season, particularly during the critical period from stem elongation to the middle grain filling, which alongside natural rainfall aims to balance the water lost due to accumulated evapotranspiration. Even so, the potential yields achieved in Spain are lower than in Chile due to a number of factors, such as Spain's shorter crop cycle duration, its higher night temperatures and the higher temperatures during the reproductive stage. The genetic advance in GY of spring durum wheat in the high-yielding environment of central Chile was 72.8 kg ha$^{-1}$ per year (0.73% per year) for the period 1964–2010, and 63.4 kg ha$^{-1}$ per year when the cultivar released in 1964 was excluded from the analysis (Figure 1). This is higher than the findings for spring bread wheat (43.5 kg ha$^{-1}$ per year or 0.51% per year) for a similar period (1964–2008) and in the same Mediterranean environment [8]. It is also clearly higher than the increase reported for durum wheat in Spain (24 kg ha$^{-1}$ y$^{-1}$; 0.44% y$^{-1}$) from 1980 to 2003, with no clear additional improvements occurring thereafter [26]. In northwest Mexico, under fully irrigated conditions, the genetic progress of spring durum and bread wheat varieties developed by CIMMYT was 0.49% and 0.41% per year, respectively, between 1966 and 2003 [27], and 0.88% per year when comparing eight bread wheat cultivars released between 1962 and 1988 [28]. A more recent study conducted at the same site in Mexico indicated that the GY progress was 30 kg ha$^{-1}$ per year (0.59%) for spring bread wheat cultivars developed from 1966 to 2019 [29]. In Spain, under moderately irrigated conditions, the genetic progress of GY was 0.36% and 0.44% for Italian and Spanish cultivars of durum wheat, respectively, for cultivars released between 1920 and 2000 [9]. In South Australia, under rain fed conditions, the annual rate of increase in GY was 25 kg ha$^{-1}$ for 13 cultivars released between 1958 and 2007 [30]. In North China, the annual genetic progress of spring bread wheat ranged from 0.48% (32.0 kg ha$^{-1}$) for cultivars released between the 1960s and the 1990s [31], and in Henan Province values of 51.3 kg ha$^{-1}$ per year have been reported for the last three decades [32].

The yield progress observed in advanced lines of the INIA-Chile breeding program (Experiment 2), which includes genetic and agronomic progress, has been much higher (569 kg ha$^{-1}$ per year) than in all the studies discussed above. This large increase in GY is explained partly by the genetic progress, but overall the improvements have derived from better agronomic management of durum wheat in the central-south of Chile, and this has included modifications to irrigation and particularly adjustments in fertilization practices conducted during the first three years of the program. In winter bread wheat, the yield progress was 246 kg ha$^{-1}$ per year (2.6%) between 1976 and 1998 in central Chile under fully irrigated conditions [14]. Clearly, fine tuning of crop management can have large impacts on GY in high-yielding environments when lines or cultivars of high yield potential are available.

Plant height was reduced from 107 cm in 1964 to an average of 90 cm for the period 1970–2010 (Figure 1A), and this was the consequence of the introduction of semi-dwarfing genes in Chile in the late 1950s [33]. Plant height was negatively correlated with the year of release in Australia, in cultivars developed between 1958 and 1973, but not in cultivars released after 1973 [30], and in China, in cultivars released between 1960 and 2000 [34]. A negative correlation between plant height and GY was also reported in the study of Zhou et al. [34]. However, the comparison of advanced lines produced during the last decade (Figure 2) showed a positive correlation between plant height and GY. These results suggest that plant height of semi-dwarf wheat below 70–80 cm may limit light interception and thus canopy photosynthesis and yield potential in high-yielding environments.

The HI increased between 1964 and 1970, but after that there were no changes. Furthermore, the correlation between GY and HI was not significant. The maximum values of harvest index (0.53) found in the current work were higher than those reported by Royo et al. [9] in a set of Italian and Spanish cultivars of durum wheat released between 1920 and 2000 and tested in Spain. In studies where cultivars released before and after the green revolution were evaluated, HI and the year of cultivar release were positively correlated (e.g., [9] for durum; [6,8,31] for bread wheat), but there was no correlation in cultivars released after 1970 (Figure 1C; see also [29]).

The increase in GY was positively associated with days to heading and KM2 and TKW (Table 3). The increase in the crop cycle in an irrigated Mediterranean environment contrasts with the breeding trend observed in rain fed Mediterranean areas, where early flowering, shorter duration cultivars are selected to escape post anthesis drought [9,35,36].

TKW increased significantly from 1964 to 1970, but the correlation with the year of cultivar release was not significant for the period 1970–2000 (Figure 1F). Genetic progress in TKW can be positive, negative or null depending on whether kernel weight has been a selection target for breeders and whether there have been changes in the number of grains per year (the trade-off between seed size and number in crops; [37]). For instance, in durum wheat growing in Mediterranean environments, kernel weight was superior in modern cultivars in Turkey [38], but remained unchanged in Italian and Spanish cultivars from the 20th century [9]. In bread wheat, kernel weight has been reduced [8,14,39] or has not changed [31] with genetic improvement.

*4.2. Grain Quality and Kernel Isotope Composition*

The test weight increased in modern cultivars and was positively correlated with TKW ($r = 0.44$; $p < 0.05$). The values of test weight obtained in this work are higher than those found in durum wheat genotypes grown under rain fed conditions in different zones of Spain [40,41]. Unfortunately, gluten and protein content did not improve between 1964 and 2010. Other studies comparing older cultivars or landraces to modern cultivars of durum wheat from Mediterranean countries have revealed lower grain nitrogen or protein content in the more modern cultivars [15,16,20,21,42]. In addition, the presence of *Rht* dwarfing genes in bread and durum wheat seems to reduce the concentration of Zn, Fe, Mn and Mg in kernels [43].

A number of studies have reported a negative correlation between grain protein concentration and GY in durum wheat [44] and in bread wheat [45,46]. It is probable that the lack of genetic progress in protein content is related to the strong increase in GY of the Chilean cultivars. However, this negative relationship should not be a limitation for genetic improvement in quality traits in grains of durum wheat because protein composition seems to be more important than the concentration [16,47].

The relationship between kernel $\delta^{13}$C and GY was negative, suggesting that genotypes exhibiting higher water use are the most productive [21,48]. In bread wheat under fully irrigated conditions, modern and more productive cultivars showed lower $\delta^{18}$O and $\delta^{13}$C, and higher stomatal conductance [8,18,21,25]. This negative relationship between $\delta^{13}$C and GY (or positive relationship between $\Delta^{13}$C and GY) has also been found in rain fed Mediterranean conditions ([21] for durum wheat; [49,50] for bread wheat), suggesting that the most productive lines are those able to maintain higher stomatal conductance and use more water [51]. In addition, the stomatal conductance of post green revolution wheat cultivars in Australia seem to show a lower sensitivity to vapor pressure deficit above 2 kPa compared to older cultivars [52], and this can be associated with lower (more negative) $\delta^{13}$C values.

In summary, changes in a number of traits have occurred in durum wheat cultivars selected for high-yielding environments in Chile. The large genetic progress in grain yield was associated with increases in days to heading, KM2 and TKW. The test weight has also increased with the year of cultivar release, but the gluten and protein content have not improved between 1964 and 2010. Interestingly, the increase in yield potential seems related to longer duration and somewhat taller plants that are able to use more water.

## 5. Conclusions

This study provided evidence that a high genetic advance in GY for durum wheat is feasible under high yielding conditions. The increase in GY was a consequence of a greater number of kernels per m$^2$ and higher kernel weight in the more modern cultivars. The test weight was lower in the 1960s and increased curvilinearly with year of cultivar release. The gluten and protein content did not improve between 1964 and 2010. GY was negatively correlated with kernel $\delta^{13}$C, suggesting that genotypes exhibiting higher water use are the most productive. The yield progress of a set of advanced lines evaluated between 2006 and 2015 was very high, due to genetic progress, but this was also due to management improvements, particularly adjustment of fertilization practices conducted during the first three years. Unlike other Mediterranean agro-environments, a longer growing cycle together with taller plants seems to be related to the increase in the GY of Chilean durum wheat during recent decades.

**Supplementary Materials:** The following are available online at http://www.mdpi.com/2073-4395/9/8/454/s1, Table S1: Monthly minimum (T min) and maximum (T max) temperatures and precipitation (PP) at Santa Rosa, Table S2: Mean values of cultivar traits according to the year of evaluation.

**Author Contributions:** I.M. designed the experiment and selected the germplasm. D.C. and K.R. were in charge of the management of the experiment and evaluation of agronomic traits. M.D.S. contributed to the isotope analysis. A.M.M.-E. contributed to data analysis. A.d.P. performed the data analysis and was in charge of writing the text but all the authors contributed to the manuscript.

**Funding:** This work was supported by the research CONICYT grant FONDECYT N° 1180252, Chile, and the contribution of Maria Dolores Serret was supported in part by the AGL2016-76527-R project from MINECO, Spain.

**Acknowledgments:** We thank Alejandro Castro for technical assistance with the field experiments.

**Conflicts of Interest:** The autors declare no conflict of interest.

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
