# Peer review of "Genetic Advance of Durum Wheat Under High Yielding Conditions: The Case of Chile"

_agronomy, doi:10.3390/agronomy9080454_

Round 1
Reviewer 1 Report
The work focuses on evaluating agronomic traits in a set of ten cultivars of durum wheat released in Chile between 1964 and 2010 and testing selected advanced lines from the Chilean breeding program from 2006 to 2015. This study provides information about the increase in yield potential and genetic progress in durum grain yield in high-yielding environment conditions. In my opinion, there is enough original information to be published. However, the manuscript needs improving, and many important aspects should be clarified before its acceptance for publication.
Specific comments:
1. Page 2, line 120: Six agronomic traits appear in M&M section. However, in table 2 and table 3, seven and eight traits were included, respectively. For example, GY and KM2 should be included in M&M section and KM2 should be included in table 2. In figure 1, only six traits were included, what about DH and KS included in Table 3? Please, include the eight traits evaluated (Table 3) in M&M section, table 2 and Figure 1. Nomenclature and description of agronomic traits evaluated should be clarify along the manuscript.
2. Since year and year x cultivar interaction is significant for almost all the traits evaluated, the mean values of the genotypes in each period should be included in the manuscript.
3. A more detailed description of the varieties/materials used in the study should be included
4. How the harvest index was computed?
5. What about the dwarfing gene Rht-B1 in the tested cultivars and advanced lines?
6. All experiment in this work have been carried out on high-yielding conditions with 3-4 irrigations per season. In the discussion section, the authors stated that values of harvest index found in this work were higher than those reported by other authors in cultivars of durum wheat tested in Spain. The authors should be indicated in the discussion which were the environmental conditions for field experiments. Had been these studies carried out on high-yielding conditions with 3-4 irrigations per season? Differences (if any) in the environmental conditions should be included in the discussion.
Reviewer 2 Report
The article entitled “Genetic advance of durum wheat under high yielding conditions: the case of Chile” sent to the Agronomy journal for publication, analyzes the changes in agronomic, quality and physiological characteristics in a set of durum wheat cultivars released in Chile between 1964 and 2010. The aim is to get some information about genetic progress in durum wheat in high yield potential environments. The number of cultivars tested is not very high (10) but they perform field experiments during three seasons.
The authors also address the progress in other traits, as yield, in advanced lines from INIA breeding program tested in several years (2006-2015). In this case, the number of lines is higher (46) and they have data from 2006 to 2015.
The manuscript could be suitable for publication, but should be improved in some aspects.
Material and methods : In the material and methods section, the plant material and field trials are well explained, but sometimes it is not clear for the reader which traits are measured in each experiment. I would suggest rewriting this section a little bit, or including a table with the traits measured in each experiment.
In the grain quality section when the gluten content method is described, I miss a reference to a previous publication or to an AACC method, just to clarify if they refers to wet gluten, total gluten….
The authors should state if for quality traits all subplots in a year from the same line were bulked or they did independent analysis.
The authors could consider adding some more information as supplementary files, i.e. a summary of the agronomic and quality data (mean, SD, min, max…).
Results: In Figure 2, the check cultivar ‘Corcolen’ is highlited and the authors say that it followed a similar trend to the advanced lines. What happened with the other three check cultivars? Did they followed also the same trend? ‘Lleuque’ is a cultivar released in 2009, was it included in the experiment since then?
In my opinion, section 3.3 should be before 3.2
Why DH are not presented in figure 1?
Discussion: The conclusions from experiment 2 are weak. All the data from this experiment are shown in Figure 2. In L253 the authors present a yield progress observed of 569 kg/ha/per year. It is not easy to get this data from Figure 2. The explanation that this progress is related to genetic progress is difficult to understand looking at the check cultivar performance. Then, in L266-269 they say that the advanced lines experiment showed a positive correlation between PH and GY, but again I am not sure if you can get this conclusion from Figure 2. The authors say in material and methods that they are only presenting trends, but this is difficult to see.
From my point of view, what is lacking in the manuscript is to justify the experiment 2, (why it is important? What it is made for?), and provide some robust conclusions from it.
Conclusions: rewriting required
Round 2
Reviewer 1 Report
Means for agronomic traits in Supplementary table S2 should be contained the standard errors or coefficient of variance by genotype and year of release. These means should be compared by appropriate statistical method (Duncan’s test or LSD test).
Even in the best years of release (2009 and 2010) is it clear from table S2 (and Fig. 1C) that any of the genotypes evaluated reached a GY mean value of 13 Mg/ha, as the authors stated in the discussion. This statement should be changed according the mean values by genotype and year of release obtained by the authors. Average values with standard errors for GY should be included in the 3.1 results section.
Please, clarify the nomenclature used in the pedigree (Table 1).
Reviewer 2 Report
The article entitled “Genetic advance of durum wheat under high yielding conditions: the case of Chile” sent to the Agronomy journal for publication, is now more understandable and suitable for publication.
Author Response
Thanks